# Effect of Attentional Focus on Sprint Performance: A Meta-Analysis

**DOI:** 10.3390/ijerph19106254

**Published:** 2022-05-20

**Authors:** Danyang Li, Liwei Zhang, Xin Yue, Daniel Memmert, Yeqin Zhang

**Affiliations:** 1School of Psychology, Beijing Sport University, Beijing 100084, China; danyangli@bsu.edu.cn (D.L.); xinyue_psy1998@163.com (X.Y.); 2Institute of Exercise Training and Sport Informatics, German Sport University Cologne, 50933 Cologne, Germany; d.memmert@dshs-koeln.de; 3China Football College, Beijing Sport University, Beijing 100084, China; yeqinzhang@bsu.edu.cn

**Keywords:** external focus, internal focus, sprint performance, meta-analysis, qualitative interaction

## Abstract

Sprinting is often seen in a variety of sports. Focusing one’s attention externally before sprinting has been demonstrated to boost sprint performance. The present study aimed to systematically review previous findings on the impact of external focus (EF), in comparison to internal focus (IF), on sprint performance. A literature search was conducted in five electronic databases (APA PsycINFO, PubMed, Scopus, SPORTDiscus, and Web of Science). A random-effects model was used to pool Hedge’s *g* with 95% confidence intervals (CIs). The meta-analysis included six studies with a total of 10 effect sizes and 166 participants. In general, the EF condition outperformed the IF condition in sprint performance (*g* = 0.279, 95% CI [0.088, 0.470], *p* = 0.004). The subgroup analysis, which should be viewed with caution, suggested that the benefits associated with the EF strategy were significant in low-skill sprinters (*g* = 0.337, 95% CI [0.032, 0.642], *p* = 0.030) but not significant in high-skill sprinters (*g* = 0.246, 95% CI [−0.042, 0.533], *p* = 0.094), although no significant difference was seen between these subgroups (*p* = 0.670). The reported gain in sprint performance due to attentional focus has practical implications for coaches and athletes, as making tiny adjustments in verbal instructions can lead to significant behavioral effects of great importance in competitive sports.

## 1. Introduction

One of the most crucial motor skills in sports is sprinting. Sprint performance enhancement is a critical component of training programs for several individual and team sports, including short-track speed skating [1], soccer [2], American football [3], and ice hockey [4]. According to the mechanics and technique, sprinting can be divided into five stages: starting, acceleration, drive phase, recovery phase, and deceleration [5]. Despite accounting for only 5% of the total 100 m race time [6], the sprint start is one of the determining components of high performance in sprinting [7], as the margins of wins in short sprints can be as nuanced as a few milliseconds. In addition, a well-executed sprint start is conducive for athletes generating sufficient acceleration, optimizing their stride length, and outperforming others when sprinting [8]. Additionally, the shorter the sprint distance, the more critical the sprint start [9]. Furthermore, athletes who perform fast sprint starts have a psychological advantage over their competitors, which can be significant in many races [10,11]. Given that sprinting performance, especially the sprint start, is a critical skill that underpins performance in many sports, there is a large amount of scientific literature on sprint training, such as plyometric intervention programs [12], resisted sled training [13,14], and lower-limb wearable resistance training [15].

Over recent years, the role of attentional focus has emerged as a significant modulator in sprint performance. Attentional focus refers to an individual’s intentional attempt to direct their attention through explicit thoughts for the purpose of executing a motor skill [16,17]. An individual can pay attention to an IF (i.e., concentration on the body movements) or an EF (i.e., concentration on the intended movement effect). For example, a coach advising a sprinter on the push phase can give an internal cue such as, “Concentrate on exploding through your hips,” or an external cue such as, “Concentrate on exploding off the ground.” Both attentional cues call attention to the power output from an explosive driving force, but the IF emphasizes the body movement (i.e., apply force with the hips) and the EF underlines the movement effect (i.e., apply force to the ground).

Although there is often a one- or two-word difference between EF (e.g., “focus on driving the ground back”) and IF (“focus on driving your legs back”) instructions, available evidence suggests that tiny adjustments in verbal cues are enough to trigger significant behavioral effects in sprinting. Performers receiving an EF direction perform better at sprinting than those receiving an IF one [18,19]. For example, Kovacs et al. [19] examined the attentional focus effect during a sprint task where participants were instructed to accelerate as fast as possible to the finish line placed 6 m from the start line. In the study, all participants finished the sprint task under the EF and IF conditions using a counterbalanced within-subjects design. The results revealed that instructing participants to concentrate on pushing the blocks away (EF) resulted in a faster start reaction relative to instructing them to concentrate on fully extending their knees (IF). The result was similar to that reported by Ille et al. [18], who found that performers displayed more efficient sprint performance when focusing on exploding from the starting blocks (EF) compared with focusing on pushing quickly with their legs (IF). In addition, relevant reviews show that the EF advantage is not only embodied in sprint tasks but also robust in other sports tasks (e.g., soccer, golf, and swimming); for reviews, see [20,21,22]. Moreover, one recent meta-analysis [23] suggests an EF exceeds IF in motor performance with a small effect size (Hedges’ *g* = 0.264).

However, there has been no comprehensive evaluation specifically on the attentional focus effect on sprint performance, but the available evidence shows the benefit of the EF strategy in sprinting. Given the findings of Porter et al. [24], a comprehensive evaluation of the experimental evidence would have significant practical use. In that study, unexpectedly, 84.6% of surveyed athletes competing at the USA Track and Field Outdoor National Championships reported their coaches utilized verbal instructions that prompted an IF during authentic athletic practice [24]. According to the field investigations, there appears to be a divergence between experimental evidence and practical applications. As we know, a systematic review is an opportunity to narrow the evidence–practice gap [25], and a meta-analytic review is characterized by its comprehensiveness, quantified evidence, and guidance for practice [26]. Therefore, this divergence calls for a systematic evaluation and meta-analysis of these experimental findings to give evidence-based recommendations and reinforce the translation of knowledge into action.

In addition to providing an overall evaluation of the impact of attentional focus on sprinting, another issue requiring an answer is whether the skill level of performers moderates the attentional focus’s effects on sprint performance. In the research field of attentional focus, there is some dispute about whether skill level interacts with the attentional focus effect. Some studies indicate that the benefits of EF instructions have generalizability and show no dependency on skill [20], but other studies have shown that novices in the early phases of skill acquisition benefit less than experts because it is difficult for novices to shift their attentional focus from focusing internally on the correct execution of skills to focusing externally on the coordination of movements [27,28]. This issue set us thinking about whether skill level mediates the attentional focus effect on sprint performance, which is probably the most fundamental movement skill. Therefore, the current study included the skill level as a moderating variable to identify whether the attentional focus effect would vary according to the performer’s skill level.

Due to the above-mentioned views, the principal objective of the present work was to systematically assess the efficacy of EF versus IF in immediate sprint performance, including start reaction time and total sprint time. The secondary objective was to compare the immediate attentional focus effect between high-skill and low-skill sprinters.

## 2. Materials and Methods

This meta-analysis conformed with the PRISMA guidelines and adhered to the pre-registration protocol on the Open Science Framework (https://osf.io/cyzu5, accessed on 29 December 2021).

### 2.1. Literature Search Strategy

Five electronic databases (APA PsycINFO, PubMed, Scopus, SPORTDiscus, and Web of Science) were searched for predetermined key terms ((“attentional focus” OR “external focus” OR “focus of attention”) AND (“sprint”)) until December 2021. Furthermore, a supplementary manual search was conducted to identify relevant articles from the reference lists of each included article. The search was restricted to peer-reviewed, English-language research articles.

### 2.2. Eligibility Criteria

Studies were considered eligible according to the following criteria:The participants were in a healthy state and over the average age of 18;An EF intervention was employed;A comparison between EF and IF was drawn;The immediate effect of focus instruction on sprint performance was examined, including start reaction time or total sprint time;A within-subjects study design was employed.

### 2.3. Literature Selection

Two reviewers (D.L. and X.Y.) independently selected the studies that met the eligibility criteria. At the end of the selection, when disagreements existed between the reviewers, the reviewers discussed the reasoning for their selections, and if one reviewer realized she or he had made a mistake, then the process was complete [29]. All cases of disagreement were solved in this way.

### 2.4. Data Extraction

One reviewer (D.L.) extracted and entered the following data from all included studies into Microsoft Excel. Another reviewer (X.Y.) checked the data for accuracy: author name (s), publication year, title, sample size, participant characteristics (age and expertise level), intervention instruction, study design, and sprint performance outcome (s).

Based on the included studies’ descriptions and the reviewers’ judgments, the participants’ skill levels were categorized into two types: high-skill and low-skill sprinters. The participants classified as highly skilled at sprinting were skilled sprinters participating in high-level competitions, collegiate track sprinters with more than five years’ experience in sprinting, or elite athletes in other sports events actively engaged in an organized strength and conditioning program including professional running form and technique training. (How long does it take to be a high-level sprinter? There is currently no unified definition. However, according to some studies’ recruiting criteria for high-level sprinters [30,31] and some studies’ reporting [32,33], it was seen as appropriate to set more than 5 years’ training experience as the cut-off for high-level sprinters. For example, Maćkała et al. [30] defined sprinters with a minimum of 5 years of training experience as high-performance sprinters, and Čoh et al. [31] regarded sprinters with at least 6 years of training experience as high-level sprinters. In addition, Ericsson et al. [32] and McAuley et al. [33] revealed that in most sporting domains, the development of high performance took 5–30 years). The participants categorized as low-skilled were novices, performers with no formal sprint mechanics training, or performers involved in other sports with no experience in sprinting training.

### 2.5. Data Synthesis

Standardized mean differences (effect sizes) were estimated using Hedge’s *g*, as it takes small sample bias into account [34]. The sample size, mean, and standard deviation under EF and IF conditions were acquired from the included studies. For the study [19] providing standard errors instead of standard deviations, standard errors were converted to standard deviations. For the studies using a within-subjects design, the actual value of the pre-post correlation coefficient was estimated from the raw data and imputed to calculate the effect size. When the pre–post correlation was unavailable, the analysis used an imputed value of 0.5 [23].

### 2.6. Data Analysis

For the primary analysis, the effect of EF versus IF on sprint performance was investigated using a random-effects model meta-analysis, taking into account the heterogeneity within attentional focus instructions, sample characteristics, and outcome measures from the included studies. The effect size was calculated using the Hedges’ *g* with 95% CIs, the magnitude of which was assessed as small (0.2~0.5), moderate (0.5~0.8), or large (>0.8) [35].

*I*^2^ was used to measure study heterogeneity, which was classified as low (25%), moderate (50%), or high (75%) [36]. The *τ*^2^ statistics are additionally reported to facilitate the interpretation of between-study heterogeneity beyond the current list of included studies [37]. In order to determine the possible source of heterogeneity, a subgroup analysis was conducted using Cochran’s *Q* test and *I*^2^ [38,39,40]. In the present study, the subgroup analysis was performed according to the participants’ skill levels to see any difference in the attentional focus effect between low-skill and high-skill sprinters. In addition to reporting whether there is a statistically significant subgroup effect (the results of the *Q* test) and the extent of heterogeneity within each subgroup (the results of *I*^2^), it is helpful to point out whether the subgroup effect is “quantitative” or “qualitative” [38,41,42] (In “quantitative interaction,” the magnitude of the treatment effect fluctuates between subgroups, but the treatment effects in different subgroups are in the same direction, whereas in “qualitative interaction,” the treatment effect is favorable in one subgroup but is unfavorable or neutral for the other subgroup [41,43,44]) in order to provide useful and practical insights for practitioners into how the focus instructions should be implemented in practice [41,43], although caution is needed when interpreting these results [45].

Funnel plot inspection [46] and Egger’s regression test [47] were carried out to examine potential publication bias. In addition, sensitivity analysis was performed by sequentially removing each study to evaluate whether the pooled estimate was influenced excessively by a single study [48].

All the statistical analyses were performed using the Comprehensive Meta-Analysis (CMA) 3.0 software (Biostat Inc., Englewood, NJ, USA).

### 2.7. Risk of Bias

Through the independent assessments of two reviewers (D.L. and X.Y.), the risk of bias of each eligible study was evaluated by the updated Cochrane Risk of Bias Tool (RoB 2.0) [49]. The evaluation considered the “effect of assignment to an intervention.” Any disagreement during the assessments was resolved by consensus. The RoB 2.0 tool designed for randomized crossover trials was applied to the studies using a within-subjects design. The RoB 2.0 tool was organized around five domains: randomization and allocation process, deviations from the intended interventions, missing outcome data, measurement of the outcome, and selection of the reported result. Each domain featured signaling questions used to assess study bias and applicability. Finally, these domain-level assessments were used to construct an overall risk of bias rating for each study.

## 3. Results

### 3.1. Included Studies

Figure 1 depicts the PRISMA flow diagram, which shows the study selection process at various stages. The search yielded 78 records from five databases. Once duplicates were deleted (*n* = 27), the abstracts (*n* = 51) and full-text articles (*n* = 11) were screened. Following the final screening procedure, the final meta-analysis comprised six studies (10 comparisons).

### 3.2. Study Characteristics

Table 1 lists the main characteristics of the included studies. Publications in this collection were published between 2013 and 2018. The sample size range varied from 8 to 84, totaling 166 participants. The participants had various skill levels, from novice to elite. The tasks included a 6 m sprint, 10 m sprint, 20 yard (18.28 m) sprint, and 20 m sprint.

### 3.3. Meta-Analytic Results

Figure 2 displays the individual and aggregate effect sizes for the effect of attentional focus on sprint performance. Overall, compared to IF, EF had a small positive impact on sprint performance (*g* = 0.279, 95% CI [0.088, 0.470], *p* = 0.004, *τ*^2^ = 0.025), demonstrating that on average, individuals who followed EF verbal instructions fared better than those who used IF. The magnitude of heterogeneity was demonstrated by an *I*^2^ value of 28.478%, which explained a moderate proportion of the between-study variance.

### 3.4. Moderator Analysis Results

Subgroup analyses were conducted based on participants’ skill levels, whether low or high (see Figure 3). The test for subgroup differences indicated that there was no statistically significant subgroup effect (*Q* = 0.181, *df* = 1, *p* = 0.670), suggesting that skill level did not moderate the effect of EF in comparison to IF.

Although no significant difference was observed between these groups, the subgroup effect was found to be “qualitative interaction” [41,44,54]. That is, the benefits of the EF strategy were significant in the subgroup of low-skill participants (*g* = 0.337, 95% CI [0.032, 0.642], *p* = 0.030, *I*^2^ = 18.527), indicating that EF instruction significantly improved their sprint performance, but the effect was not significant among high-skill participants (*g* = 0.246, 95% CI [−0.042, 0.533], *p* = 0.094, *I*^2^ = 43.223), indicating that EF did not significantly improve their sprint performance. However, caution should be exercised when considering these results (see Discussion).

### 3.5. Publication Bias and Sensitivity Analysis

A visual inspection of the funnel plot (see Figure 4) showed that the distribution of the included studies was slightly asymmetrical. However, Egger’s regression test (*p* = 0.222) showed no statistically significant publication bias. In addition, the results of the sensitivity analysis revealed that the pooled estimate of the attentional focus effect on sprint performance ranged from *g* = 0.227 (95% CI [0.052, 0.402]) to *g* = 0.317 (95% CI [0.118, 0.516]) after one study was removed at a time, indicating that no single study considerably changed the overall estimate.

### 3.6. Risk of Bias

Table 1 shows the results of the risk of bias assessments. All eligible studies were scored as causing some concerns. The most dominant problem was the randomization process, which caused some concern in all studies, partly because of the inadequate reporting of allocation concealment.

## 4. Discussion

The present study aimed to use a meta-analytic statistical technique to assess the effect of an EF on sprint performance. The findings from the main analysis demonstrated that employing EF had a small positive effect on sprint performance compared to IF (*g* = 0.279, *p* = 0.004) but also revealed moderate heterogeneity in the data (*I*^2^ = 28.478%). Additionally, the subgroup analysis suggested that the benefits of the EF strategy were significant among low-skill sprinters (*g* = 0.337, *p* = 0.030) but not significant among high-skill sprinters (*g* = 0.246, *p* = 0.094), although no significant difference was seen between these subgroups (*p* = 0.670). Together, these findings show that EF outperforms IF in sprint performance.

### 4.1. Why Is EF Better for Sprint Performance Than IF (i.e., Attentional Focus Effect)?

The constrained action hypothesis [55,56] can explain this phenomenon. That is, when attention is focused externally, it permits the motor control system to work under nonconscious automatic processes, resulting in reflexive movement and improved performance. Conversely, as attention is focused internally, the motor control system is constrained by consciously regulated processes, resulting in less reflexive and reduced fluent movement patterns. Consequently, IF generated worse outcomes than EF. Over the past two decades or more, myriad studies have lent support to this hypothesis and have revealed the generalizability of EF’s benefits, as shown over a broad range of motor tasks (e.g., balance, dart throwing, soccer, ballet, golf, running, swimming, and jumping), populations (at various skill levels, ages, and disabilities), and performance measures (movement accuracy, movement form and movement efficiency); for reviews, see [20,21,22].

In addition, Wulf and Lewthwaite [57,58] recently proposed the goal-action coupling mechanism in the OPTIMAL theory, illustrating the neuromuscular change under EF circumstances, which could provide another explanation for the benefits of EF. That is, like the other two factors (enhanced expectancies and autonomy support), EF improves neuromuscular efficiency [59,60,61]. The enhanced neuromuscular efficiency, to a certain extent, indicates the establishment of smooth, sizable, and opportune functional connectivity between task-relevant motor networks with less effort, and the disconnection of unnecessary linkages relatively easily [57]. Furthermore, an EF, which is helpful in suppressing needless neural activity [62] and muscular co-contractions [59], contributes to clarifying neuromuscular coordination. Sprinting, as a complex motor skill, requires the contraction of several muscle groups at accurate times and intensities during the stride cycle for the purpose of achieving optimal sprint performance [16]. Therefore, through more efficient muscle activation at the neuromuscular level, an EF has the possibility to boost sprint performance.

### 4.2. How Should We Interpret the Subgroup Analysis of the Attentional Focus Effect for High-Skill Sprinters and Low-Skill Sprinters?

First, the subgroup analysis shows that skill level does not play a moderating role in the attentional focus effect. However, the uneven covariate distribution (six effect sizes in the high-skill group vs. four effect sizes in the low-skill group) suggests that the subgroup analysis may be incapable of yielding accurate results [42], and a smaller number of participants contributed data to the high-skill subgroup (sample sizes of 57) than to the low-skill subgroup (sample sizes of 109) means that the analysis may be unable to identify subgroup differences [42]. It would be advisable for more research to be undertaken in the research field to examine the subgroup effect.

Second, although no significant difference was seen between the low- and high-skill subgroups, a qualitative interaction was found between the attentional focus effect and the skill level. Although some researchers suggested that qualitative interaction of that kind should be considered “with skepticism” [45], other researchers deemed that the qualitative interaction may be more frequent [41] and more important [43] in terms of public health and clinical implications because the pathways between intervention and outcome are frequently intricate, and the treatments may have varying effects on various populations through different pathways, and consequently, these results could help policymakers or practitioners develop effective and refined intervention programs. In order to provide practical implications for sports practitioners, the current study has performed further analyses of the qualitative interaction, although caution is needed when interpreting these results.

That is, on the one hand, the benefits in favor of the EF strategy were significant in the subgroup of low-skill participants. However, due to the number of effect sizes included in the analysis being small (i.e., four effect sizes in the low-skill group), fewer than the five recommended by Jackson and Turner [63], the analysis does not have enough statistical power to conclude that EF intervention has a powerful effect on low-skill performers. Careful investigation via more randomized controlled trials may provide stronger evidence, which would be useful to suggest the direction of future research.On the other hand, the benefits of EF instruction were non-significant in the subgroup of high-skill participants. There are three possible explanations for this observation. First, the compensation account (“who has will less be given”), which is used to explain the improvements in cognition from cognitive training, primarily in working memory and attention control processes (e.g., [64]), could shed light on the aforementioned result. That is, the highly skilled individuals already perform close to the ceiling and cannot improve as much. Therefore, baseline performance might be negatively related to intervention gains. Another possible explanation for this is, as Wulf [65] put forward, that for attentional focus effects to occur, a certain level of task challenge appears to be needed. If the task has already been automated, there will be a lower possibility for the attentional focus effect to emerge. Obviously, it appears that the sprint task is not difficult enough for the highly skilled participants because the sprint task is already handled with a high degree of automaticity due to previous practice and experience; thus, directing attention externally has no further benefit. Moreover, an alternative explanation, mentioned in the study of Winkelman et al. [53], is also plausible—namely, the way that abstracting information or meaning from attentional focus instructions is influenced by the level of experience. Abstracting information or meaning involves the ability to overlook unnecessary elements while focusing on the critical components and constructing generalized thoughts based on previous experience and knowledge [66]. According to one study [67], individuals with a lot of experience usually extract and summarize the general subjective implication from given instructions. As a consequence, highly skilled performers may frequently return to their natural focus, which has become consolidated and automated after years of practice, and self-select the appropriate goal-relevant aspect to focus on irrespective of the instructions presented. Therefore, the difference between EF and IF instructions may be canceled out among the high-skill performers. However, more randomized controlled trials are needed to investigate the true effect. In addition, the distance effect of EF in the facilitation of motor performance and learning has been found in several studies [68,69]. That is, increasing the distance of an EF from the body movement improves sports performance and motor learning, and the distance effect seems to be more effective in high-skill performers [70]. Future studies may take the distance of EF into consideration to improve high-skill sprinters’ performance.

Finally, as Higgins et al. [38] emphasized, it must be remembered that subgroup analyses are essentially observational and are not performed on the basis of randomized comparisons. Hence, subgroup analyses have the limitations inherent in every observational study, such as potential bias due to confounding with other study-level factors. Therefore, caution is needed when interpreting findings from subgroup analyses [40].

### 4.3. Applied Practice and Future Directions

This study highlights that sprinting coaches and athletes should be aware of the advantage of EF and induce or adopt EF during practice and competitions to boost sprint performance in competitive venues. As shown in this study, making minimal changes in verbal instructions can result in considerable performance improvement. Yet, it is worth noting that what seems to be missing from focus instruction literature is the examination of the effect of attentional focus instructions in competitive settings. Although Martin et al. [71] pointed out the complexity of competitions and the difficulty of controlling limited sport psychological interventions in competitions, well-designed research investigating the attentional focus effect on sprint competitions will further advance the application of attentional focus. Furthermore, based on the study of Winkelman [17], there are three characteristics of attentional focus cues (i.e., distance, direction, and description) that moderate the positive influence of an EF. Therefore, coaches employing attentional focus intervention are encouraged to take the distance, direction, and description of an EF into consideration and provide athletes with individually tailored EF instructions. In addition, it would be useful to examine any potential changes in the intervention’s efficacy based on the type of sports event (e.g., individual sport versus team sport), and the type of motor action. Most importantly, according to Porter et al. [24], there is some inconsistency between science and application in giving verbal instructions to athletes in track and field sports. Sport scientists need to bridge this gap by reaching out to coaches and athletes, giving an ear to their practical concerns, and conveying scientific methods to optimize sports performance.

### 4.4. Limitations

When evaluating the study’s findings, the following limitations should be kept in mind. First, the limited number of included studies imposed restrictions on assessing the effect of attentional focus in sprinting. Furthermore, most of the studies included had a small sample size, which could result in overestimating the attentional focus effect in relation to studies with larger sample sizes. Therefore, when examining and reproducing the efficacy of attentional focus interventions in sprint performance, larger sample sizes should be used to increase statistical power. In addition, the inadequate descriptions of participants’ skill levels in the included studies made it difficult to rate the actual skill levels of participants; therefore, future studies should provide clarified descriptions of participants’ expertise.

## 5. Conclusions

In summary, the present meta-analysis corroborates that EF could improve sprint performance compared to IF. Furthermore, a qualitative interaction has been found between the attentional focus effect and skill level, although the subgroup analysis showed that skill level does not play a moderating role. Specifically, the attentional focus effects are significant in low-skill sprint performers but non-significant in high-skill sprint performers. However, caution is needed when interpreting the subgroup analysis. Finally, coaches and athletes themselves should be aware of the strength of EF and design EF strategies in accordance with their skill development to improve sprinting, as a 1% improvement can increase the chances of success and boost medal positions in international competitions, just like it did for Lyles, who beat Coleman by a scant 0.006 s (9.852 to 9.858) at the finish line in the 2019 Diamond League Shanghai 100 m.

## Figures and Tables

**Figure 1 ijerph-19-06254-f001:**
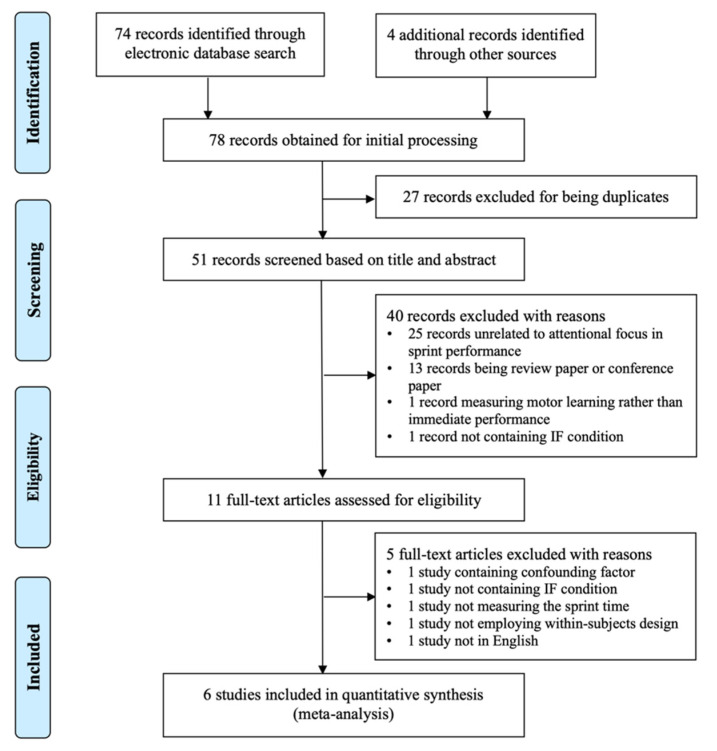
Flow diagram of literature selection process.

**Figure 2 ijerph-19-06254-f002:**
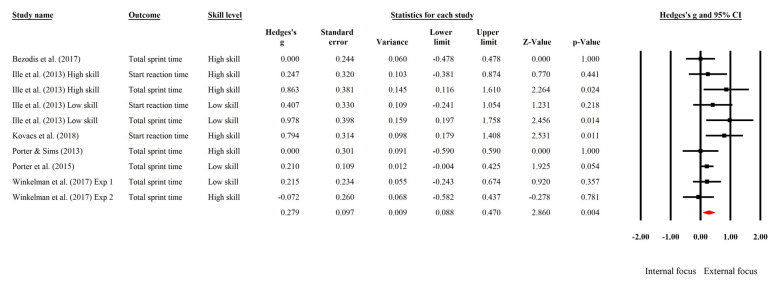
Forest plot of effect sizes in sprint performance studies comparing external focus versus internal focus [18,19,50,51,52,53].

**Figure 3 ijerph-19-06254-f003:**
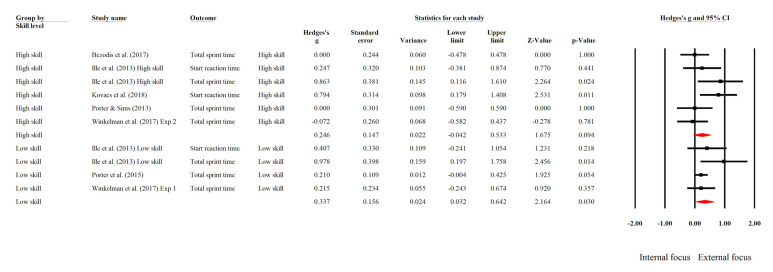
Forest plot of subgroup analysis for comparing the attentional focus effect on sprint performance between high-skill and low-skill sprinters [18,19,50,51,52,53].

**Figure 4 ijerph-19-06254-f004:**
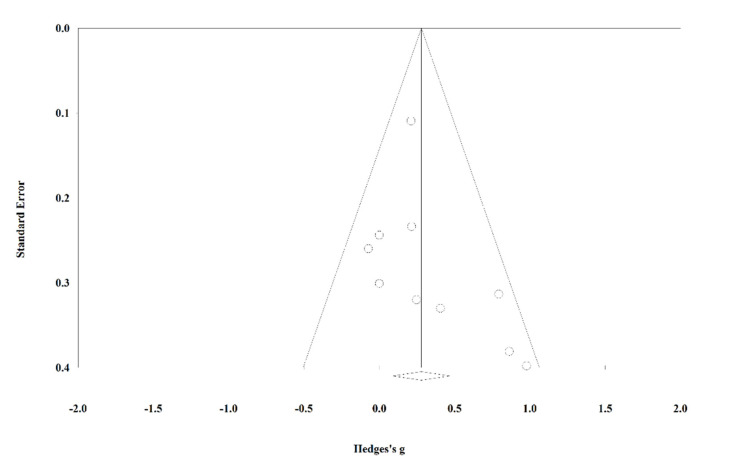
Funnel plot of effect sizes of sprint performance studies comparing external focus and internal focus.

**Table 1 ijerph-19-06254-t001:** Summary of the included studies’ characteristics.

Study	Year	Skill Level	Sample Size	Age (Year)	Task	Verbal Instruction	Outcome	Overall Bias
Bezodis et al. [50]	2017	high-skill	15	22 ± 4	10 m sprint	EF: “focus on clawing backwards at the ground with your shoe in every step you take” IF: “focus on pulling your leg backwards just before each contact with the ground”	total sprint time	some concern
Ille et al. [18]	2013	high-skill	8	20~30	10 m sprint	EF: “get off the starting blocks as quickly as possible, head towards the finish line rapidly, and cross it as soon as possible” IF: “push quickly on your legs and keep going as fast as possible while swinging both arms back and forth and raising rapidly your knees”	total sprint time; start reaction time	some concern
		low-skill	8					some concern
Kovacs et al. [19]	2018	high-skill	12	20.8 ± 1.7	6 m sprint	EF: “focus on pushing the blocks away” IF: “focus on extending your knees”	start reaction time	some concern
Porter and Sims [51]	2013	high-skill	9	21.11 ± 1.22	20-yard sprint	EF: “while you are running the 20-yard dash with maximum effort, focus on gradually raising up. Also, focus on powerfully driving forward while clawing the floor as quickly as possible” IF: “while you are running the 20-yard dash with maximum effort, focus on gradually raising your body level. Also, focus on powerfully driving one leg forward while moving your other leg and foot down and back as quickly as possible”	total sprint time	some concern
Porter et al. [52]	2015	low-skill	84	20.32 ± 1.73	20 m sprint	EF: “while you are running the 20-m dash, focus on driving forward as powerfully as possible while clawing the floor with your shoe as quickly as possible as you accelerate” IF: “while you are running the 20-m dash, focus on driving one leg forward as powerfully as possible while moving your other leg and foot down and back as quickly as possible as you accelerate”	total sprint time	some concern
Winkelman et al. [53]	2017 Exp 1	low-skill	17	19.41 ± 1.06	10 m sprint	EF: “focus on driving the ground back as explosively as you can” IF: “focus on driving your legs back as explosively as you can”	total sprint time	some concern
	2017 Exp 2	high-skill	13	28 ± 4.32	10 m sprint		total sprint time	some concern

## Data Availability

The data used for the meta-analysis are available from the corresponding author upon request.

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
