# Peer review of "Effect of Attentional Focus on Sprint Performance: A Meta-Analysis"

_ijerph, 2022, doi:10.3390/ijerph19106254_

Round 1

Reviewer 1 Report

Dear Autors,

the research paper presented for review has been well planned and its conclusions answer important issues related to sprint. In the work, clear goals were set and appropriate research tools were used, the analysis was carried out in accordance with the methodology, and interesting, practical conclusions refer to the presented problem.
Particularly valuable for the results of the work is: "Literature Selection: Two reviewers (D.-Y.L. and X.Y.) independently selected the studies that met the el-102 igibility criteria. At the end of the selection, disagreements were discussed between the 103 reviewers, and a consensus was reached."

Reviewer 2 Report

Please see the review file.

Reviewer 3 Report

Considerations and comments in attached document

Round 2

Reviewer 2 Report

General

The authors did a highly commendable job in addressing all the concerns raised in the previous round of review. The introduction of the concept of qualitative interaction, including the pros and cons of its use, in assisting readers in interpreting the subgroup analysis results is especially commendable. Other contextual elaborations have also improved the relevance of content in this revised version of the manuscript. The footnotes are informative in providing the technical clarity required for a better understanding of important methodological considerations. The reporting of publication bias results is adequate.

Minor Comments

  1. 2 Lines 63–64: It may be more appropriate to use “faster” to refer to “reaction” because “shorter” is more commonly used with noun words of direct reference to time.
  2. 2 Line 95: Do you mean “different” instead of “differential”?
  3. 9 Lines 290–291: Consider rephrasing to “sample size of 57” and “sample size of 109” because a sample is usually used as a collective noun to refer to the entire group of participants.
  4. Pg. 9 Lines 307–308: The assessment of statistical power is based on the number of effect sizes (minimum of five as recommended by Jackson & Turner, 2017) that can be extracted from all included studies, rather than the number of trials in these studies.
